# Effect of Irrigation on Sugarcane Morphophysiology in the Brazilian Cerrado

**DOI:** 10.3390/plants13070937

**Published:** 2024-03-23

**Authors:** Laryssa Maria Teles Batista, Walter Quadros Ribeiro Junior, Maria Lucrecia Gerosa Ramos, Vinicius Bof Bufon, Rodrigo Zuketta Sousa, Christina Cleo Vinson, Sidnei Deuner

**Affiliations:** 1Faculty of Agronomy and Veterinary Medicine, University of Brasília, Brasília 70910-900, Brazil; laryssatelles@yahoo.com.br (L.M.T.B.); ccvinson@gmail.com (C.C.V.); 2Brazilian Agricultural Research Corporation—(EMBRAPA Cerrados), Planaltina 73310-970, Brazil; rodrigo.zuketta@gmail.com; 3Brazilian Agricultural Research Corporation—(EMBRAPA Meio Ambiente), Jaguariuna 13918-110, Brazil; vinicius.buffon@embrapa.br; 4Botany Department, Biology Institute, Federal University of Pelotas, Pelotas 96010-900, Brazil; sdeuner@yahoo.com.br

**Keywords:** *Saccharum* spp. L., water deficit, biomass production, gas exchange, industrial quality

## Abstract

Since sugarcane is semi-perennial, it has no escape from water stresses in the Brazilian Cerrado, and consequently, drought impacts plant growth and industrial quality. The objective of this study was to evaluate the morphophysiology and quality of the first ratoon of two sugarcane varieties submitted to irrigated and stressed treatments under field conditions. For the biometric characteristics, in general, significant decreases were observed under the stressed treatment for all periods, and only minor differences were detected between the studied cultivars. Physiological parameters decreased under stressed conditions, but to a different extent between the varieties. RB855536 was able to maintain a greater rate of transpiration. Productivity was reduced by 103 t ha^−1^ for variety RB855536 and 121 t ha^−1^ for RB867515, compared to plants with full irrigation during the dry period, but cane quality was similar in both genotypes. Measurements of physiological and morphological parameters may prove useful in the rapid identification of genotypes with greater tolerance to abiotic stress.

## 1. Introduction

Sugarcane is an economically important crop that is used as a raw material for sugar and alcohol production and can also be used to generate electricity from the burning of bagasse as well as be used in animal and human feed [1]. The current forecast for sugar production in the 2023/2024 harvest is 468 million tons, and the total ethanol production is estimated to be 60 billion liters [2]. The increase in the demand for sugarcane production occurred due to the need for clean energy generation and the Brazilian sugar alcohol industry is expanding, occupying areas where sugarcane production was previously uncommon, and due to the climatic characteristic of the Brazilian Cerrado, it is subject to a long period of water deficit. Sugarcane is one of the most efficient crops for producing ethanol in terms of water footprint (WF), representing the amount of water consumed per unit of ethanol produced [3,4].

Sugarcane shows a reduction in growth in certain periods due to the adverse winter weather conditions of the Cerrado in the Brazilian central–western region, such as lower water availability. The most accentuated development phase begins with the resumption of regular rainfall, an increase in light intensity and temperature, and occurs from December to March [5]. In the central–west region of Brazil, even in regions with long dry periods, it is customary to make a single 40/60 mm rescue irrigation right after harvest, to guarantee sugarcane sprouting [6]. Therefore, irrigation plays an important role in sugarcane production, especially in regions with a long drought period, like the Cerrado [7,8]. Rodolfo Junior et al. [9] found higher sugarcane yields during the dry period in the Brazilian Cerrado when the water applied exceeded the amount commonly used by the farmers.

In the period between autumn and winter, when there is a decrease in rainfall and lower temperatures, there is greater maturation activity and less growth activity, with intense sugar storage. Carvalho et al. [10] found that a 75% replacement of evapotranspiration in the Cerrado region is sufficient to express the productive capacity of sugarcane. Identifying morphophysiological characteristics linked to drought tolerance would give us a hint of what parameters we could evaluate in a breeding process during the dry period.

The availability of water and temperature are factors that greatly affect the growth of the crop and vary according to the different phenological stages of sugarcane [11,12]. Studying the growth patterns of different sugarcane varieties is important so that the phases of maximum development coincide with periods of greater water availability and solar radiation, allowing the culture to express its full productive potential [13]. The study of morphological characteristics of sugarcane, biomass production, and its partitioning and dynamics provides information regarding the mechanisms that increase the crop’s productive efficiency in response to environmental conditions [13]. In addition, plant growth and yield under water stress are related to a reduced photosynthetic rate, transpiration, and stomatal conductance [14]. During water stress, leaf area photosynthesis and sugarcane yield can be reduced by 38%, 56%, and 48%, respectively [15]. The identification of the productive capacity of different varieties of sugarcane and the investigation of the effects of management on the culture is generally carried out through the analysis of growth and the evaluation of some morphological variables of the plants, such as height, number of plants per meter, leaf area, and productivity [16]. The growth dynamics of the stalk is a variable that shows a positive correlation with the final yield of sugarcane [17] and occurs as a function of the variety and the environmental conditions [18]. The knowledge of the relationship between morphophysiological, productive, and industrial quality characteristics in conditions of low water availability helps to create criteria for selecting genetic materials that are more responsive to these limiting conditions.

The objective of this study was to evaluate the morphophysiology and quality of the first ratoon of two sugarcane varieties submitted to irrigated and stressed treatments under field conditions.

## 2. Results

### 2.1. Water Stress and Sugarcane Varieties’ Development

The mean stalk diameter (MSD) and mean stature of the stalks (MSS) were influenced by the water regime for both varieties, showing that adequate water supply through full irrigation leads to greater increases in the diameter and stature of the stalks (except 167 DAC for RB867515 in Figure 1A and 296 DAC for RB867515 in Figure 1B). For the stressed treatment, there was no significant difference in MSD between the varieties during the drought period (80 to 147 DAC, Figure 1A). Regular precipitation resumed at the beginning of October and point 167 DAC was performed near the end of October, where precipitation of approximately 259 mm had already occurred. MSD evaluations from 167 DAC to 296 DAC show that RB867515 presented a higher recovery and the differences between the varieties were occurring progressively (Figure 1A), such that the resumption of regular rainfall resulted in a significant compensatory growth in MSD for RB867515 (Figure 1A). For the control (fully irrigated treatment), there was no difference between both studied varieties (Figure 1A), for except 99 and 259 DAC. For the MSS analysis, in the fully irrigated treatment, the genotype RB855536 presented greater stature of the stalks in all evaluation points (except 201 DAC, Figure 1B). The stressed treatment had a similar height and the recovery of regular precipitation (points 167 to 296) promoted a greater increase in the stature of the stalks in RB867515. At 296 DAC, the difference in height between the fully irrigated varieties was 48.1 cm, in which RB855536 presented 321.07 cm of MSS and RB867515 showed 272.93 cm, and there was no difference between the stressed and fully irrigated treatments for RB867515 but still a significant difference for RB855536 (Figure 1B).

Leaf length +3 (C + 3) was generally greater in the fully irrigated treatment, with plants obtaining a larger photosynthetically active leaf area, showing that the water supply influenced this parameter. There was a small difference at the measurements’ start (point 80). Significant differences were observed concerning the water regime during the drought period (points 99 to 147), and when regular precipitation was resumed (points 167 to 259), these differences decreased. By the last evaluation point, there were no differences in leaf length for both varieties (296 DAC, Figure 2A). However, this parameter cannot distinguish between both studied varieties as they had similar values in stressed and fully irrigated conditions (except 201 DAC and 127 DAC, respectively, Figure 2A). Leaf width +3 (L + 3) increased during the period of the experiment, but variety RB867515 showed no significant differences among treatments (except 201 and 259 DAC, Figure 2B), while RB855536 was significant (except 127 DAC, Figure 2B). Under the stressed treatment, RB867515 maintained a greater leaf width for most points, while under full irrigation, there was no significant difference (except 99 DAC, Figure 2B).

The number of open green leaves (NOGL) analysis showed more open green leaves in the fully irrigated treatment than in the stressed treatment (points 80 to 147, Figure 3A). After regular precipitation was resumed (points 167 to 296 DAC), there were no differences between treatments for the RB867515 variety, but fully irrigated RB855536 had more open leaves compared to the stressed treatment and the RB867515 variety (Figure 3A).

The number of emergent leaves (ELs) did not show significant results in relation to the water regime (except 167 DAC, Appendix A) and when comparing the varieties in each water regime (except 201 DAC, Appendix A). The number of dead leaves (DLs) increased as the experiment was carried out; however, there were no significant differences for most points between the fully irrigated and stressed treatments for and between both varieties (Appendix A). The number of tillers decreased as the experiment was carried out, but no significant differences were observed in relation to the number of tillers between the water regimes (except 147 DAC, Appendix A) and between varieties (Appendix A).

Leaf area index (LAI) evaluations were performed at 86, 99, 145, 179, 218, and 275 DAC. Except for the evaluation performed at 86 DAC, for the variety RB855536, the varieties RB867515 and RB855536 that were irrigated presented a higher LAI compared to the plants submitted to the water stress (Figure 3). In addition, for the water stress treatment, in general, the LAI of RB855536 was greater than that of RB867515. After the resumption of precipitation, in the evaluation performed at 179 DAC in the water stress treatment, the LAI showed higher values but still had significant differences between water regimes.

### 2.2. Leaf Water Potential and Gas Exchange in Two Sugarcane Cultivars under Water Stress

Leaf water potential (Ψw) showed a significant reduction in plants submitted to water restriction (stressed treatment) when compared to those that were fully irrigated (Figure 4). A comparison of the studied varieties showed that with stress at 86 DAC, there was no significant difference between the varieties only in the rainfed treatment, but when the stress period was extended to 113 and 133 DAC, the two varieties presented a distinct trend, where RB855536 was significantly more sensitive to water deficit (Figure 4). Comparing genotypes only in the fully irrigated treatment, for 86 DAC, RB855536 had a higher water potential. In the first evaluation, at 86 DAC, the variety RB855536 presented a Ψw that was 18.2% lower in the plants from the stressed treatment than the full irrigation, whereas, for the variety RB867515, that reduction was 43.7%. At 133 DAC the RB855536 had a greater reduction, reaching values around 100% lower in the stressed treatment plants than the fully irrigated plants (Figure 4).

The responses of the sugarcane varieties in gas exchanges (Figure 5) are in agreement with the results observed for leaf Ψw (Figure 4). The transpiration rate of both varieties was significantly higher in the fully irrigated treatment than in the stressed treatment in the three periods evaluated (Figure 5A). For the fully irrigated treatment, water application management allowed the plants to maintain greater transpiration since they did not suffer stress and thus kept their stomata open. Transpiration at 133 DAC under both the fully irrigated and stressed treatments was lower than at 86 and 113 DAC. Even with these relatively lower transpiration values, the difference between the treatments in the last period was significantly more pronounced due to the extension of the water stress experienced by the plants in the stressed treatment, as was also observed in leaf Ψw (Figure 4). Thus, the variety RB855536 presented a reduction in transpiration rate of 4.7 mmol m^−2^ s^−1^ in the fully irrigated treatment to 1.6 mmol m^−2^ s^−1^ in the stressed treatment. In contrast, for RB867515, the difference was less expressive: 2.8 mmol m^−2^ s^−1^ in the fully irrigated treatment and 1.6 mmol m^−2^ s^−1^ in the stressed treatment. Although RB855536 presented higher transpiration values than RB867515 in 86 and 113 DAC for the stressed treatment, there was no significant difference at 133 DAC because both varieties had very low transpiration rates.

The stomatal conductance (Figure 5B) followed a similar behavior to that observed for transpiration. There was a significant difference between the treatments in the three evaluated periods for the two sugarcane varieties, with lower values for the plants kept under stress and with the increase in the stress period, where the difference in the stomatal conductance between the stressed and fully irrigated plants became more accentuated. At 113 DAC, the study varieties presented significantly similar values, and at 133 DAC there was no difference between varieties for the stressed treatment as both had very low stomatal conductance, and for the fully irrigated treatment, it was higher for RB855536.

For the photosynthetic rate (Figure 5C), no differences between treatments at 86 DAC were observed, with RB855536 showing higher values. For the following periods, the two varieties suffered a significant reduction in photosynthesis in the stressed treatment. At 133 DAC, the fully irrigated variety RB855536 presented a rate of 13.7 µmol m^−2^ s^−1^ compared to 3.0 µmol m^−2^ s^−1^ for the stressed treatment, and 9.2 μmol m^−2^ s^−1^ in the fully irrigated treatment and 3.4 μmol m^−2^ s^−1^ in the stressed treatment for RB867515.

In the evaluation of the chlorophyll index, significantly lower values were observed in the plants in the stressed treatment; however, this parameter was not efficient for distinguishing the varieties according to the water regime (Figure 6).

For the evaluation of the leaf area index (LAI), only the first period (86 DAC) coincided with the physiological analyses, and the following two evaluations were performed at 99 and 145 DAC (Figure 7). Water stress promoted a lower LAI for both varieties in the three periods studied; it increased as stress was applied (except at 86 DAC, RB855536, Figure 7), and under fully irrigated conditions, the two varieties presented a similar LAI (except at 99 DAC, Figure 7). The variety RB855536 presented higher values of LAI compared to RB867515 in the stressed treatment at 86 and 145 DAC.

### 2.3. Productivity and Quality

The yield of stalks was affected by a water deficit, and with full irrigation, it was possible to reach higher yields for both varieties. However, there was no statistical difference between the studied varieties for either water regime. In the yield in tons of stalks per hectare (TCH), the fully irrigated treatment produced 230 tons ha^−1^ for both varieties, and in the stressed treatment, this was reduced to 127 tons ha^−1^ and 122 tons ha^−1^ for RB855536 and RB867515, respectively (Figure 7).

The evaluation of Brix and broth purity (Figure 8A,C) did not show significant differences in relation to the water regime and between the varieties. The broth pol (Figure 8B) showed significant differences with respect to the fully irrigated water regime but not between varieties, with percentages of apparent sucrose (Pol) of 14.54 and 14.11% for the fully irrigated treatment and 13.59% and 13.15% for the stressed treatment for RB855536 and RB867515, respectively. The fiber content in the fully irrigated and stressed treatment for RB855536 was 12.89% and 12.31%, respectively, and 12.05% and 11.9% for the full irrigation and stressed treatment for RB867515, respectively. Fiber percentage did not present significant differences in relation to the water regime, with only a slightly significant difference between varieties in the fully irrigated treatment (Figure 8D). The cane pol presented significant differences with respect to the water regime only in RB867515, and no significant differences were observed between varieties (Figure 8E). The values of recoverable total sugars (TRS) showed a significant difference between treatments but not among varieties, with 121.37 and 115.69 kg ATR/tonne of sugarcane in the full irrigation and stressed treatment for RB867515, respectively, and 119.76 and 112.87 kg TRS/tonne of sugarcane in the fully irrigated and stressed treatment for RB855536, respectively (Figure 8F).

## 3. Discussion

### 3.1. Drought Impacts Growth and Morphology in Both Genotypes

A lack of water may affect sugarcane stalk morphology, with stalk diameter and plant height both influenced by water availability [16,19]. Moreover, the measurement of such morphological parameters has been proposed as a rapid method for assessing productivity under a water deficit in breeding programs [20]. Here, in both genotypes, drought led to reductions in stalk length and diameter. There were, however, differences between the genotypes for these parameters. While stalk diameter was similar under fully irrigated conditions, under drought RB867515 exhibited a greater diameter from 167 DAC. On the other hand, RB855536 had longer stalks under full irrigation, a difference that was abolished under drought. Stalk morphology, therefore, responded differently to drought in the two genotypes despite the absence of differences in productivity; this response was not observed in a drip irrigation study [21], perhaps reflecting the difference in irrigation strategy.

Water stress often causes marked leaf senescence and restriction in the emergence of new leaves and leaf curling, which is a mechanism to tolerate the reduction in available water [22]. There are also alterations in the development of the architecture of the vegetative canopy of sugarcane, which is fundamental for crop productivity since it intercepts the solar radiation that, in turn, acts in the processes of photosynthesis and transpiration [22]. Both RB855536 and RB867515 exhibited reduced leaf length and width, leaf area index, and number of open green leaves under drought at one or more time points. While differences in the leaf length between the genotypes were minimal, drought tended to have a greater effect on leaf width in RB855536 than in RB867515. The number of open green leaves tended to be greater in RB855536 than RB867515 under fully irrigated conditions, but under drought, the situation inverted at 127 and 147 DAC when the number for RB867515 was greater. On the other hand, the number of emergent leaves (ELs) did not indicate a trend in the behavior of the varieties as was verified for the other biometric parameters; thus, ELs cannot be considered as an indicator of water stress in or tolerance of the varieties.

The results indicate that both varieties are responsive to the adequate water supply because when rainfall reestablishment occurred, it presented a higher NOGL. When submitted to full irrigation, the NOGL was also high. However, during the period of intense water restriction, RB867515 maintained its NOGL, which may indicate drought tolerance. The fully irrigated treatment had a significantly higher LAI than the stressed treatment even after precipitation resumption at 179 DAC (except point 86 DAC for variety RB855536, Figure 3). In general, both varieties performed similarly in both treatments. The increase in LAI occurs due to the higher number of leaves and the increase in the individual leaf area. There is a significant association between crop productivity and total photosynthetically active surface.

Given that we detected no differences in productivity between the two genotypes, it may be that differences in leaf gas exchange were at least partially offset by canopy alterations, but this hypothesis requires further testing.

### 3.2. Leaf Water Potential and Gas Exchange in Two Sugarcane Cultivars under Water Stress

In addition to productivity and industrial quality, other morphophysiological parameters can help understand the drought tolerance and responsiveness to irrigation, facilitating the understanding of tolerance mechanisms, and avoiding working with a black box in the genotype selection process for this purpose. An overexpression of genetically modified sugarcane genes linked to drought has been validated under field conditions [23]. However, differences among conventional genotypes are known. RB867515 and RB855536 exhibit different physiological responses to drought. Stomatal closure is one of the main mechanisms to prevent dehydration under limited water availability, reducing transpiration [24]. Such responses may be rapid, with stomatal closure leading to reduced CO_2_ supply for photosynthesis and reduced transpiration, which affect thermal energy dissipation and nutrient transport by mass flow [25]. The sensitivity of this mechanism can vary considerably between the sugarcane genotypes, and the ability for genotypes to maintain stomata that are open for longer under water-limited conditions may represent greater tolerance or resistance to this stress [16,26]. While the yield of both varieties was the same under drought, physiological parameters were impacted to differing extents. As expected, drought reduced the leaf water potential, but this occurred to a greater extent for RB855536 than for RB867515. Such changes can occur rapidly, as only four days of suspension of irrigation were required to produce a decrease of more than 1 MPa in predawn leaf water potential for RB867515 [27].

Lower leaf water potential might reflect a greater osmotic adjustment in RB855536 that would facilitate transpiration maintenance. Indeed, RB855536 maintained a higher transpiration rate than RB867515 under both the fully irrigated and stressed treatments for 86 DAC and 113 DAC. Sugarcane is known to accumulate osmolytes under osmotic stress [28,29,30]. A combined omics analysis of the same two varieties grown under greenhouse conditions indicated that both RB867515 and RB855536 undergo osmotic adjustment during drought stress, both through the accumulation of sugars and the release of amino acids during protein hydrolysis [26]. This study suggested a greater accumulation of sugars, including the osmoprotectant sugar raffinose in RB867515 compared to RB855536. It is worth noting that here, the varieties were analyzed at the same leaf water potential and under controlled conditions. Similar differences to those detected for transpiration were also observed for stomatal conductance and net photosynthesis at 86 DAC, though not at 113 DAC where these parameters were only greater in RB855536 for the fully irrigated condition. Basnayake et al. [31] also found a correlation between stomatal conductance and sugarcane yield.

A relatively sensitive response in RB867515 was also reported in a previous study, where stomatal conductance decreased more rapidly in this genotype than in RB962962 [27]. Despite the physiological similarities between drought and salt stress, whilst 48 days of salt stress led to reduced net photosynthesis and leaf water potential in both genotypes, no difference was detected between them [25]. Bezerra et al. [29] also suggested comparing RB867515 and RB855536 and, using silicon treatment, obtained an increase in proline, superoxide dismutase, and ascorbate peroxidase in stressed plants. Other authors showed that RB867515 is a drought-tolerant variety and showed proline accumulation as a drought mechanism [32,33], especially in severe water stress [34].

Trentin et al. [35] obtained that in RB867515, the daily transpiration rate of sugarcane under conditions of severe water stress (−1500 < Ψ < −1100 kPa) was approximately 73% lower compared to plants under full water availability. In addition, the water deficit reduced the transpiration of the plants.

The different responses of net photosynthesis to drought in the two varieties do not appear to derive from an effect of drought on chlorophyll abundance, as drought resulted in decreases in chlorophyll index but with no differences between the two varieties. A similar effect was observed in a study comparing RB867515 and RB962962 [27]. Proteomic, phosphoproteomic, and transcriptomic analyses of RB867515 and RB855536 under drought stress revealed differences between the responses of these two genotypes to a water deficit, including those related directly to photosynthesis and reactive oxygen species detoxification [26]. Further investigation will be required to determine whether such alterations may be responsible for the physiological differences detected here. Both the chlorophyll index and gas exchange parameters have been used as indirect selection criteria for sugarcane genotypes [36], meaning that despite the lack of difference in productivity between RB855536 and RB867515 in our study, the identification of alterations in these physiological parameters may be useful for breeding programs aimed at increasing drought tolerance.

### 3.3. Effects of Drought on Productivity and Sugarcane Quality

Water stress during sugarcane development can result in decreased cell division and expansion, which causes a decrease in dry matter accumulation, a lower shoot growth rate and leaf area index, and lower sucrose concentration in shoots [22]. Consequently, a water deficit can reduce sugarcane productivity by up to 60% [36]. Here, we investigated the effect of growth under drought (stressed) conditions on the yield, cane quality, and physiological parameters of two varieties (RB867515 and RB855536). Both of these varieties, which share a parental line (RB72454), were within the ten most cultivated in Brazil during 2017/2018 [37] with RB867515 being the most cultivated and second most planted.

As anticipated, drought led to a significant reduction in productivity for both RB867515 and RB855536. A similar genetic origin may be one reason for small differences found between the genotypes. Silverio et al. [38] found RB867515 to be more drought tolerant than RB855536 (considering the number of tillers, specific leaf mass, chlorophyll fluorescence, stomatal conductance, chlorophyll content, and relative water content in the leaf). On the other hand, RB855536 was superior under optimal water conditions. In our study, full sprinkler irrigation using the line source method resulted in greater productivity than drip irrigation methods used by Gava et al. [21] (140 t ha^−1^ and 122 t ha^−1^ for RB867515 and RB855536, respectively). According to Conab [39] in the 2011/2012 crop season, the average TRS was 138 kg per ton of sugarcane, so the TRS values obtained here are below that expected for this harvest. However, this could be due to differences in technological and industrial processing.

Despite RB867515 being described as a drought-tolerant variety [26,33] and reports of greater root mass in pot-grown plants of RB867515 compared to RB855536 [26], we detected no differences in productivity between the varieties under either fully irrigated or stressed conditions. Differently, [40] found differences among genotypes for drought tolerance conducted in a greenhouse. These results mirror a previous report [21], though a further study identified greater stalk and sugar productivity for RB867515 under drought conditions compared to RB855536 [39]. On the other hand, Liu et al. [41] and Zhao et al. [42] compared the same group of genotypes and found a high genetic correlation between well-watered and water-limited conditions. In addition to decreasing productivity, drought led to small but significant decreases in total recoverable sugars, broth pol, and cane pol. However, once again, no differences were detected between the varieties except for a small difference in fiber content.

The fiber proportion in sugarcane is around 14% [43], and these values are similar to those obtained for both cultivars, although RB867515 showed a lower fiber content than RB855536 in the irrigated treatment (Figure 8). The varieties with a higher percentage of fiber have a greater resistance to tipping and, in general, present a greater resistance to the penetration of pests. For the industry, the fiber content is important for burning in the boilers, generating steam that will be transformed into electric energy [44]. Fiber content is also interesting for second-generation energy cane [43,45]. Energy-high fiber cane has the ability to maintain higher stomatal conductance (gs) than traditional ones at the same soil water status level, even though leaf water potential was similar between the genotypes [46].

A comparison of the same varieties under drought and drip irrigation detected no influence of water availability on cane pol [21], similar to that observed for all quality criteria when comparing plants fully irrigated with 75% of the reference evapotranspiration to those grown under drought [9]. Absolute values of quality parameters were similar to those previously reported [9,21], indicating the suitability of the varieties for industrial use when grown under fully irrigated and stressed conditions.

## 4. Materials and Methods

### 4.1. Site Description and Experimental Design

The experiment was located at EMBRAPA Cerrados (15°35′30″ S, 47°42′30″ W, altitude of 1000 m). The average annual rainfall is 1395.6 mm, and with a 21.9 °C average yearly temperature. According to Köppen, the climate is classified as tropical seasonal (Aw) with two defined seasons: drought and rainy [47]. The soil of the experimental area is a clayed Oxisol [48].

Before the installation of the experiment, a chemical analysis at 0–20 layer was performed, with pH H_2_O: 5.08; Al: 0.39 cmolc dm^−3^; P: 0.22 mg dm^−3^; K: 8.0 mg dm^−3^; Ca: 0.56 cmolc dm^−3^; Mg: 0.26 cmolc dm^−3^; H + Al: 3.7 cmolc dm^−3^; and organic matter: 0.87%.

Three months before and at the establishment of the experiment, 4 t ha^−1^ of lime and 500 kg ha^−1^ of gypsum plus 50 kg ha^−1^ of FTE BR-10 (2.5% B, 1.0% Cu, 4.0% Mn, 0.1% Mo, 4.0% Fe, 7.0% Zn, and 0.1% Co), were applied, respectively. Sugarcane was initially fertilized, with 600 kg ha^−1^ of NPK (4-30-16), and FTE BR-10.

Sugarcane was planted manually in June 2010, using two varieties that belong to a research consortium called Ridesa [37] led by public universities, RB867515 and RB855536, both of medium maturity. RB867515 is a fast-growing variety, adapted to low-fertility and sandy soils, with water restrictions and morphological characteristics including erect growth, medium tillering, good sprout budding, lodging resistance, poplar age, and medium width roots without aerial rooting. In addition, it has high agricultural productivity with high sucrose content and resistance to common and orange rust [37]. The variety RB855536 presents good clumping, erect purplish-colored stalks, and semi-open sheaths. It has high industrial productivity and good bouquet budding. Ideally, the variety should be planted in high-fertility environments with an adequate water supply.

Initially, the experiment received homogeneous irrigation during the dry period (May to October), until the first cut (11 months after the planting date), with repositioning of evapotranspiration.

The first top-dressing fertilization was applied manually and, after the first cut of the plants, was carried out in two periods: 400 kg ha^−1^ NPK (using formula 4:30:16) at the same time as rescue irrigation and 600 kg ha^−1^ of NPK in November (18 months after the planting date) at the beginning of the rainy season.

After the first cut and subsequent sprouting, the experiment was conducted in a randomized block with three replications in split-plot design; each variety constituted a main plot, which was subdivided into water regimes (WRs) (full irrigation and stressed, I and S, respectively). The experimental unit was composed of a subplot (water regime). The subplot area was formed by a row of plants 4 m long and 1.5 m wide, discarding 0.5 m on each side to avoid the effect of the border.

The line source methodology [49] was adapted in the irrigation bar as in [50]. Sprinklers with decreasing sizes were used from the central area of the bar to the end of the bar, promoting a gradient of water, as shown in Figure 9.

The WRs described before were obtained using a 20 m wide sprinkler irrigation bar (Irriga Brasil model 36/42), connected to a self-propelled TurboMaq 75/GB, with adjustable speed according to the water level to be applied (Figure 9). The full irrigation management at the highest water level was carried out, with 100% replacement of the evapotranspiration potential of the crop, estimated through the climatological balance, using the meteorological station near the experimental area. The stress treatment was carried out with rescue irrigation during the dry period. This treatment received only 60 mm of water. Note that rainfall was not observed in the experimental area from May to October (Figure 10).

Water was applied through a self-propelled system with an irrigation bar with conjugated nozzles and XI-Wobbler emitters. Watering was performed at intervals of 15 days, totalling 415.7 mm at the end of the evaluation period, whereas for the rescue irrigation treatment, the water supply was completely suspended to reproduce the reality of the production systems of some plants in the central-western Brazilian region, which has cane fields implanted in areas under long periods of water deficiency due to rainfall seasonality.

### 4.2. Biometric Evaluations

For the biometric evaluations, in both varieties, five stalks were randomly identified in the area of each subplot, such that 15 representative stalks of each treatment were chosen. The biometric evaluations were performed at 80, 99, 127, 147, 167, 201, 259, and 296 days after cutting (DAC). The following biometric parameters were evaluated: number of industrializable tillers (NIT), mean stalk diameter (MSD), mean stem stature (MSS), number of open green leaves (NOGL), emergent (ELs) and dead (DLs) leaves, length (C + 3) and width (L + 3) of the +3 leaf.

The leaf denominated as +3 was used for the evaluation of leaf dimensions, according to the Kuijper system, since it is completely developed from the physiological point of view and totally unfolded morphologically. The NIT was obtained by counting tillers along the useful area of the plot, any shoot formed from the planted cane being considered a tiller, including the primary stalk. The MSD was measured at the base of the stalks with the aid of a digital caliper. The MSS was measured using a scale, measuring from the soil level to +1 leaf, defined according to the Kuijper system as the first fully developed leaf with a visible ligule. NOGL, ELs, and DLs were quantified in each of the selected tillers. NOGL and DLs were obtained by counting all leaves completely developed from leaf +1, according to the leaf identification of the Kuijper system. Green leaves were considered as those that possessed 50% of the leaf blade visually green. The number of ELs was determined considering the leaves above leaf +1 that were not fully expanded. The L + 3 and C + 3 data were obtained by the measurements in the median portion and in the extreme points of the +3 leaves. Leaf area index (LAI) was determined using the SunScan equipment (SunScan Canopy Analysis System SS1.1M, DELTA-T) at 86, 99, 145, 179, 218, and 275 DAC.

### 4.3. Physiological Evaluations

The two varieties were evaluated for their physiological responses: leaf water potential (Ψw), photosynthetic parameters (net photosynthesis, stomatal conductance, and transpiration), and chlorophyll index evaluated at 86, 113, and 133 DAC. The leaf area index (LAI) was evaluated at 86, 99, and 145 DAC. For the parameters analyzed, five stalks of each variety were selected randomly from each subplot, using the third visible ligule leaf according to the Kuijper scale as leaf +3. Gas exchange was performed between 8 and 11 a.m. with an IRGA infrared gas analyzer (Brand ADC, Model LC Pro S/32662) with a CO_2_ concentration of 400 ppm. Each evaluation was performed in +3 leaf and 15 evaluations were obtained for each replication.

The Ψw foliar was evaluated in the pre-dawn, with a device determining the water stability of the plants (Scholander pump, model 3005, Soil). The chlorophyll index was evaluated using a digital chlorophyll meter (SOIL CONTROL, CFL-1030). The leaf area index (LAI) was determined using a SunScan (SunScan Canopy Analysis System SS1, UM.1M, DELTA-T).

### 4.4. Productivity and Quality of Stalks

The harvest of the experiment was performed manually at 337 DAC. The plot area was harvested, sorted, counted, and weighed to determine productivity, expressed in tonnes of stalks per hectare (TCH). The weights of the stalks were determined in the field with a scale (Kern HCB Model 99K50, variation of 100 g). Before the total sampling of the experiment, seven stalks were randomly sampled in each sub-plot for the analysis of the technological indexes that define the quality of the sugarcane as a raw material.

The following parameters were evaluated: pol of broth, Brix, fiber, purity, pol of cane, and total recoverable sugars (TRS), in a period less than 36 h after harvest. Brix expresses the percentage of soluble solids contained in the sugarcane juice. Pol in the broth represents the percentage of sucrose contained in a solution of sugars, whereas sugarcane contains three types of sugars: sucrose, glucose, and fructose. From the pol of the broth, the pol of the cane can be determined when using the fiber content of the cane (%). Fiber is the water-insoluble matter contained in the cane. Purity is the percentage of sucrose contained in the soluble solids of the broth. Total recoverable sugars (TRS) are the total reducing sugars recovered from sugarcane to syrup and are expressed in kilograms per ton of sugarcane.

The data were analysed by the statistical program SAS version 9.4 (Statistical Analysis System Institute, Cary, NC, USA), with a significance level of 5%, and the Tukey test, at 5% probability, was used to compare the means.

## 5. Conclusions

Irrigation caused large differences in productivity for both genotypes, but cane quality was maintained. Drought had different effects on morphology and gas exchange in the two genotypes; however, this did not translate into differences in productivity. Measurements of physiological and morphological parameters may prove useful in the rapid identification of genotypes with a greater tolerance to abiotic stress.

## Figures and Tables

**Figure 1 plants-13-00937-f001:**
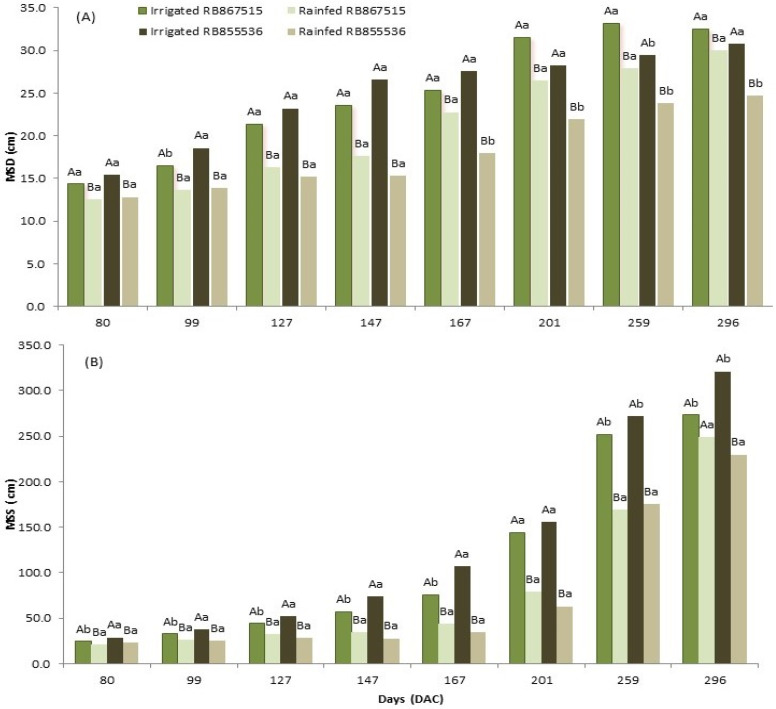
(**A**) Mean stalk diameter (MSD) and (**B**) mean stalk stature (MSS) of two sugarcane varieties (RB867515 and RB855536) evaluated at 80, 99, 127, 147, 167, 201, 259, and 296 days after cutting (DAC), under the water regime: full irrigation and stressed treatment. In each evaluation period, capital letters compare the water regimes for each variety and lowercase letters compare the varieties within each water regime. Means followed by equal letters do not differ in a Tukey’s test at 5% probability.

**Figure 2 plants-13-00937-f002:**
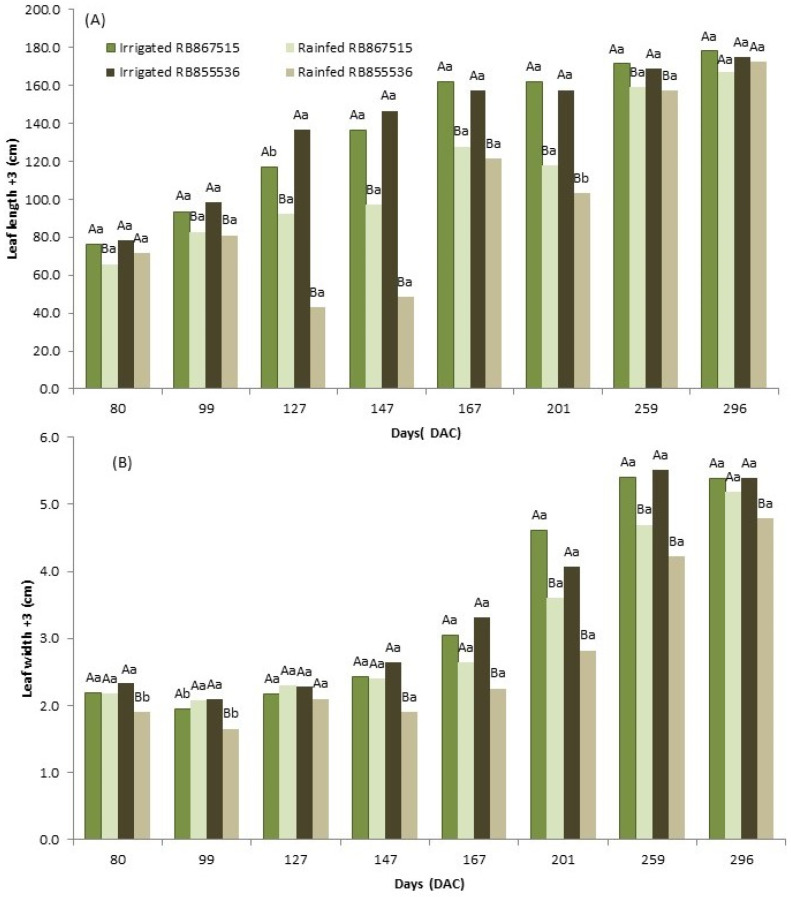
(**A**) Leaf length +3 (C + 3) and (**B**) leaf width +3 (L + 3) of two sugarcane varieties (RB867515 and RB855536) rated at 80, 99, 127, 147, 167, 201, 259, and 296 days after cutting (DAC) under the water regimes: fully irrigated and stressed. In each evaluation period, capital letters compare the water regimes for each variety and lowercase letters compare the varieties within each water regime. Means followed by equal letters do not differ in a Tukey’s test at 5% probability.

**Figure 3 plants-13-00937-f003:**
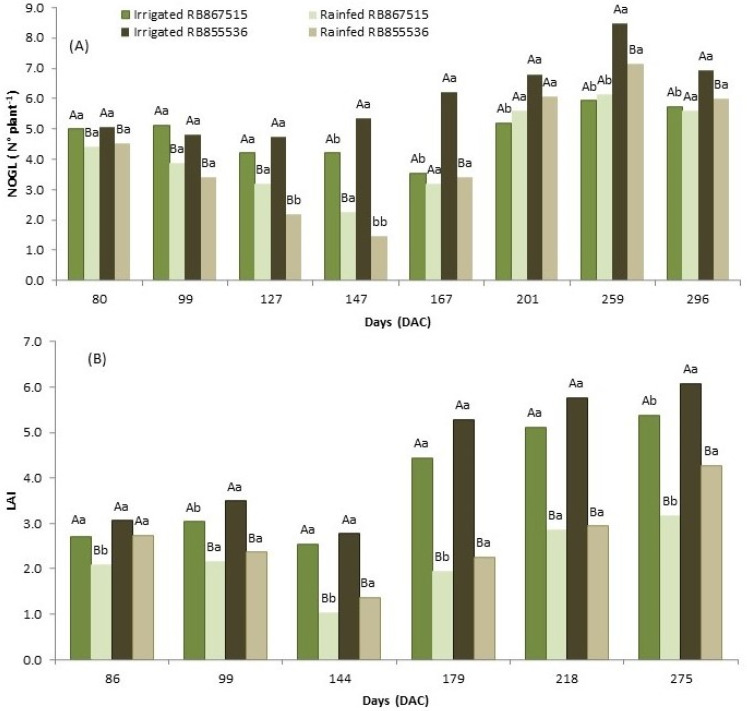
(**A**) Number of open green leaves (NOGL) evaluated at 80, 99, 127, 147, 167, 201, 259, and 296 days after cutting (DAC) and (**B**) leaf area index (LAI) evaluated at 86, 99, 145, 179, 218, and 275 days after cutting (DAC) under the water regimes of two sugarcane varieties (RB867515 and RB855536): fully irrigated and stressed. In each evaluation period, capital letters compare the water regimes for each variety and lowercase letters compare the varieties within each water regime.

**Figure 4 plants-13-00937-f004:**
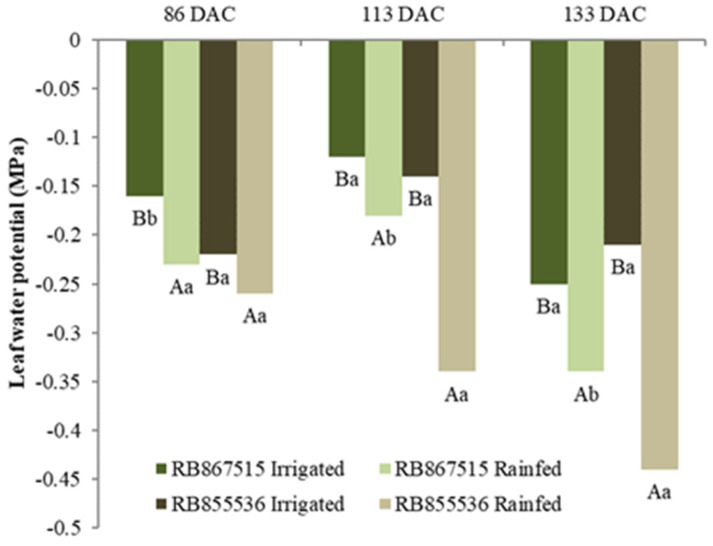
Leaf water potential of two sugarcane varieties (RB867515 and RB855536) evaluated at 86, 113, and 133 days after cutting (DAC), cultivated under water regimes: full irrigation and stressed. In each evaluation period, uppercase letters compare the water regimes for each variety, and lowercase letters compare the varieties within each water regime. Means followed by equal letters do not differ in a Tukey test at 5% probability.

**Figure 5 plants-13-00937-f005:**
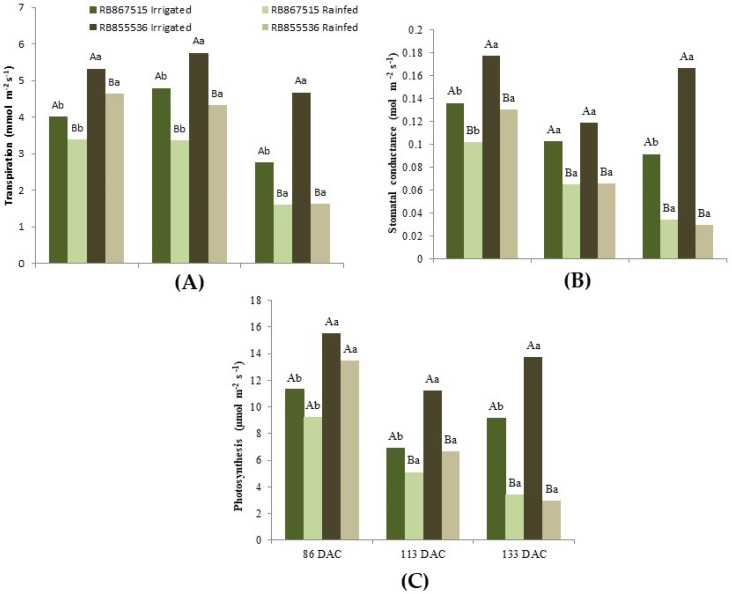
Transpiration (**A**), stomatal conductance (**B**), and liquid photosynthesis (**C**) of two sugarcane varieties (RB867515 and RB855536) evaluated at 86, 113, and 133 days after cutting water regimes: fully irrigated and stressed. In each evaluation season, uppercase letters compare the water regimes for each variety, and lowercase letters compare the cultivars within each water regime. Means followed by equal letters do not differ in a Tukey test at 5% probability.

**Figure 6 plants-13-00937-f006:**
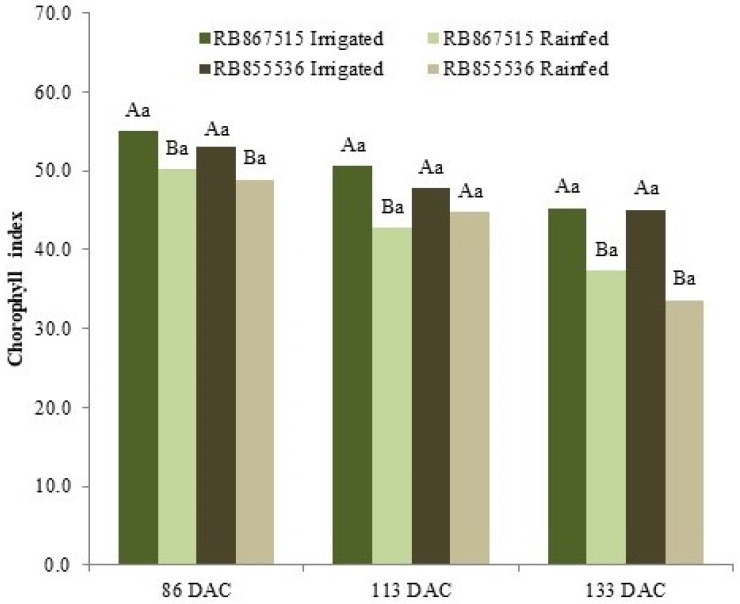
Chlorophyll index of two varieties of sugarcane (RB867515 and RB855536) evaluated at 86, 113, and 133 days after cutting (DAC), cultivated under water regimes: fully irrigated and stressed. In each evaluation period, uppercase letters compare the water regimes for each variety and lowercase letters compare the varieties within each water regime. Means followed by equal letters do not differ in a Tukey test at 5% probability.

**Figure 7 plants-13-00937-f007:**
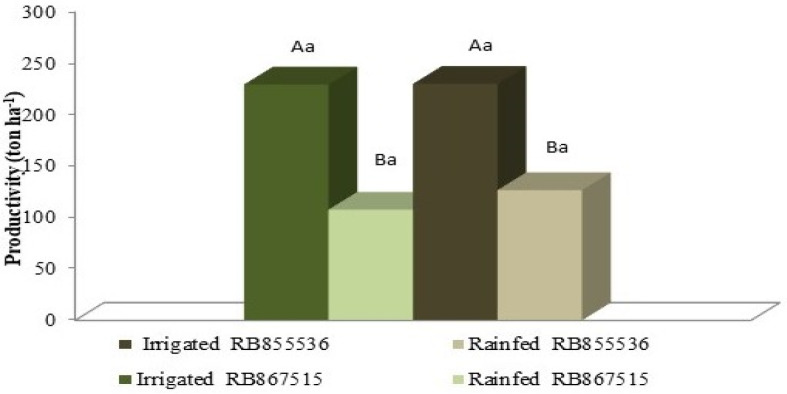
Productivity of two sugarcane varieties (RB855536 and RB867515) cultivated under water regimes: full irrigation and stressed. Capital letters compare the water regimes for each variety, and lowercase letters compare the varieties within each water regime. Means followed by equal letters do not differ in a Tukey’s test at 5% probability.

**Figure 8 plants-13-00937-f008:**
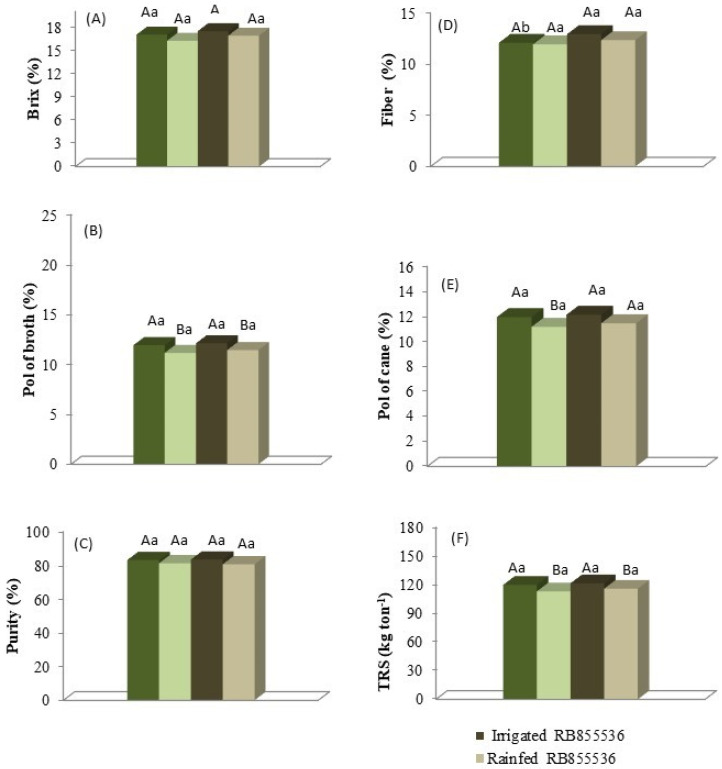
(**A**) Brix, (**B**) Pol, (**C**) broth purity, (**D**) fiber, (**E**) sugarcane pol, and (**F**) total recoverable sugar content (TRS) of two sugarcane varieties (RB867515 and RB855536) under the water regimes: fully irrigated and stressed. Capital letters compare the water regimes for each variety, and lowercase letters compare the varieties within each water regime. Means followed by equal letters do not differ in a Tukey’s test at 5% probability.

**Figure 9 plants-13-00937-f009:**
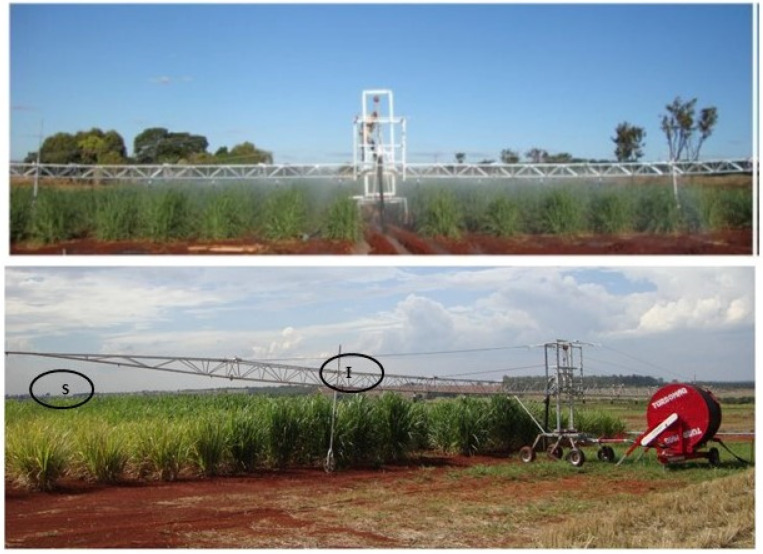
Image showing the irrigation bar and treatments stressed (S) and irrigated (I).

**Figure 10 plants-13-00937-f010:**
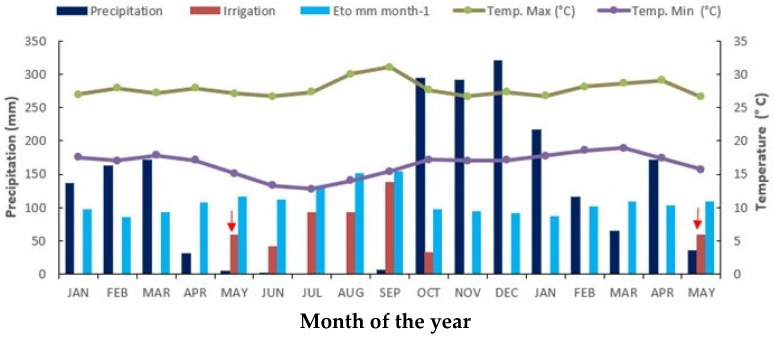
Precipitation, evapotranspiration, irrigation of sugarcane cultivars with 100% reposition of evapotranspiration (irrigated sugarcane), and mean and maximum temperature from January 2011 to May 2012. Data were collected from the Experimental Station of EMBRAPA Cerrados. 
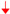
 Indicate rescue irrigation in irrigated and water stress treatments.

## Data Availability

The data presented in this study are openly available in http://repositorio2.unb.br/jspui/handle/10482/13566, accessed on 2 January 2024.

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
