# Peer review of "Effect of Irrigation on Sugarcane Morphophysiology in the Brazilian Cerrado"

_plants, 2024, doi:10.3390/plants13070937_

Round 1

Reviewer 1 Report

Comments and Suggestions for Authors

The article presents the impact of irrigation on sugar cane from a regional perspective.

The article meets the requirements of an original scientific work and deserves to be published.

Comments:

1. Be sure to change the order of the chapters. The methodological chapter should be placed after the introduction and before the results.

2. Consider changing the title: The effect of irrigation on sugarcane morphophysiology in the Brazilian Cerrado.

3. Line 552: correct conclusion - irrigation caused large differences in productivity (see Figure 7)

4. In Figure 10, show the irrigation doses in months.

Author Response

Reviewer 1

The article presents the impact of irrigation on sugar cane from a regional perspective.

The article meets the requirements of an original scientific work and deserves to be published.

Comments:

  1. Be sure to change the order of the chapters. The methodological chapter should be placed after the introduction and before the results.

Thank you for this observation, but we followed the rules of Plants, as results are presented before the methodology.

  1. Consider changing the title: The effect of irrigation on sugarcane morpho physiology in the Brazilian Cerrado.

We corrected but only removed the word “The” at the beginning of the title.

  1. Line 552: correct conclusion - irrigation caused large differences in productivity (see Figure 7)

We corrected

  1. In Figure 10, show the irrigation doses in months.

We included data from irrigation during the dry season

Reviewer 2 Report

Comments and Suggestions for Authors

This paper deals with the investigation of the effects of drought stress treatment on sugarcane biological performance in two varieties. In experimental design, the field test was very nice because the authors clearly observed the differences between control- and drought-treated samples. In addition, the differences of some parameters related to growth were also elucidated in the two varieties. I understand field tests are hard work and valuable for science community. I have a few minor comments.

1) I cannot follow the statistical analysis. Why the authors simply compare among 4 samples (RB867515 irrigated,RB855536 irrigated, RB867515 Rainfed and RB855536 Rainfed)?

2) I am glad that the authors show SD bars in each diagram.

3) In this field trial, heat stress did not occurred and there were no symptoms that indicate the effects? 

Comments on the Quality of English Language

Minor editing of English language is required. 

Author Response

Reviewer 2

This paper deals with the investigation of the effects of drought stress treatment on sugarcane biological performance in two varieties. In experimental design, the field test was very nice because the authors clearly observed the differences between control- and drought-treated samples. In addition, the differences of some parameters related to growth were also elucidated in the two varieties. I understand field tests are hard work and valuable for science community. I have a few minor comments.

  • I cannot follow the statistical analysis. Why the authors simply compare among 4 samples (RB867515 irrigated, RB855536 irrigated, RB867515 Rainfed and RB855536 Rainfed)?

This is because we decided to compare the extreme treatments.

  • I am glad that the authors show SD bars in each diagram.

As each graph has so much information, and we used a rigorous test (Tukey´s test) to compare the means, we thought there would be too much information in each graph.

  • In this field trial, heat stress did not occurred and there were no symptoms that indicate the effects? 

Sugarcane is adapted to heat. Considering that the yield increases with irrigation,  it means that water is the main limiting factor.